# A Bibliometric Analysis on Research Progress of Earthworms in Soil Ecosystems

**DOI:** 10.3390/biology13060385

**Published:** 2024-05-28

**Authors:** Jiayi Chen, Shufang Chen, Ziqiang Liu, Lizhu Wu, Huimin Xiang, Jiaen Zhang, Hui Wei

**Affiliations:** 1Department of Ecology, College of Natural Resources and Environment, South China Agricultural University, Guangzhou 510642, China; kayeechan5114@163.com (J.C.); 18818300019@163.com (S.C.); liuziqiang0201@163.com (Z.L.); wulizhu@biozeron.com (L.W.); rabbitxhm@163.com (H.X.); 2Guangdong Laboratory for Lingnan Modern Agriculture, South China Agricultural University, Guangzhou 510642, China; 3Guangdong Engineering Technology Research Center of Modern Eco-Agriculture and Circular Agriculture, South China Agricultural University, Guangzhou 510642, China; 4Key Laboratory of Agro-Environment in the Tropics, Ministry of Agriculture and Rural Affairs, South China Agricultural University, Guangzhou 510642, China

**Keywords:** earthworm, soil, CiteSpace, HistCite, bibliometrics

## Abstract

**Simple Summary:**

A bibliometric analysis of studies on the topic of earthworm in the soil ecosystem was conducted to reveal the research advances and trends in the field. The ecological effects of earthworms, the impact of agricultural activities on earthworms, earthworm ecotoxicology and earthworm invasion were four research hotspots on the topic and “impact”, ‘’biodiversity”, “oxidative stress”, “diversity”, “response”, “*Eisenia fetida*” and “exposure” have been the emerging active topics.

**Abstract:**

The earthworm, as a soil engineer, plays highly important roles in the soil ecosystem for shaping soil structure, promoting soil fertility, regulating microbial community composition and activities and decomposing soil pollutants. However, the research progresses on this important soil fauna have rarely been reviewed so far. Therefore, we conducted a bibliometric analysis of the literature published during 1900–2022, which was collected from the Web of Science Core Collection (WoS). The results showed that three periods (1900–1990, 1991–2005 and 2006–2022) could be identified in terms of the intensity of publications on the topic, and the number of publications kept increasing since 2006. The United States produced the highest publication record at the country scale, whereas Chinese Academy of Sciences was the most productive institution. Chinese institutions and authors played an active and prominent role during 2018–2022. *Soil Biology & Biochemistry* was the most popular journal for the topic-related research. In these publications, Professor Lavelle P was the most influential author. Based on a citation network of the top 50 cited papers, four hotspots were identified, i.e., the ecological effects of earthworms, the impact of agricultural activities on earthworms, earthworm ecotoxicology and earthworm invasion. Moreover, “impact”, “biodiversity”, “oxidative stress”, “diversity”, “response”, “*Eisenia fetida*” and “exposure” were the emerging and active topics in recent years. This study can help us to better understand the relevant subject categories, journals, countries, institutions, authors and articles and identify the research hotspots and emerging trends in the field of soil earthworm research.

## 1. Introduction

Soil is a vital component of the terrestrial ecosystem, as the pedosphere intersect with the atmosphere, hydrosphere, biosphere and lithosphere, and supports human health, food security and the agricultural economy [1]. However, global soil health is facing threat from soil acidification, salinization, climate change and large amounts of toxic substances, such as heavy metals and pesticides [2], which further affects the sustainable development of agriculture and food security. With accelerated urbanization and industrialization, for instance, soil heavy metal pollution becomes more and more severe and widespread [3], which changes the soil microbial community [4], affects crop growth [5], harms the agricultural environment [6] and consequently threatens the health of humans [7]. Soil acidification is a serious global environment problem in agriculture production [8] and the acidified soil area continues to expand due to the massive discharge of acid-causing substances [9], irrational use of chemical fertilizers [10] and intensive agricultural production [11]. These serious pollutions alter soil physicochemical properties and quality, limit nutrient supplies to plants, affect microbial community structure and interactively alter the biological activities and metabolic process of soil animals.

Earthworms, as the ecosystem engineer, are one of the most important soil fauna in terrestrial ecosystems [12]. As estimated, there are more than 4000 species of earthworms that belong to 12 families and 181 genera [13], which could be divided into three ecological types, i.e., epigeic, endogeic and anecic, according to the feeding habits and ecological functions [12]. They can modify soil structure and fertility, enhance soil organic matter content, speed up nutrient cycling, promote plant growth and improve soil properties by means of swallowing, digesting, excreting and burrowing in their habitats. Earthworm cast increases the aggregate stability and proportion of macroaggregates, likely due to the cementation of soil aggregates by high Ca in the soil [14,15]. The mucus secretion by earthworms’ guts is also beneficial for the formation of macroaggregates [16,17]. Christopher et al. showed that earthworms improved soil moisture and fertility and enhanced soil porosity to promote plant biomass by swallowing, excreting and burrowing [18]. The contents of total soil N and organic carbon were increased because of the decomposition and transformation of leaf litter by earthworms, thereby plants and microorganisms could obtain nutrients more easily [19]. Gong et al. [20] and Chen et al. [21] pointed out that earthworms affected the soil buffering capacity through changing soil aggregates and pH values and regulating the composition of the soil microbial community. Vermicompost has a high acid buffering capacity and therefore can improve the ability of soil to resist effects of exogenous acids [22,23]. Moreover, earthworms can work collaboratively with other soil organisms to promote ecosystem services. Compared with control without earthworms, the burrows of earthworms are beneficial for increasing the abundance, diversity and activity of the soil microbial community and consequently accelerating the decomposition of soil organic matter [24,25]. Therefore, it is of great significance to protect and utilize earthworms for improving soil environment and agriculture sustainable development.

In 1881, Charles Darwin highlighted the importance of earthworms and proposed that earthworms were highly important for soil formation and development. In spite of the high importance of earthworms, however, there are few reviews to completely outline the research advances in this field, leaving us unaware of the latest research hotspots and emerging topics. The existing literature often highlights a certain topic at a limited time scale. For instance, Lemtiri et al. [26] focused on the impacts of earthworms on soil components and dynamics, and Xiang et al. [27] analyzed the earthworm research worldwide from 2000 to 2015. In this study, we made a comprehensive summary of research on the topic of earthworms in the soil ecosystem from 1900 to 2022 by using CiteSpace and HitsCite software for bibliometric analyses. We mainly aimed to address three questions as follows: (1) Which subjects were the most popular areas in earthworm studies? (2) Which countries, institutions and authors made the greatest contribution to this field, and how about the collaboration network? (3) What are the emerging hotspots in soil and earthworm research?

## 2. Materials and Methods

### 2.1. Literature Collection

In this study, we collected literature in the database of Web of Science (WoS) Core Collection, which covered Science Citation Index Expanded (SCI-E) and Social Science Citation Index (SSCI). A topic search of “soil” and “earthworm” from the title, abstract, keywords and keyword plus of publications was conducted on 18 October 2022 to collect literature in the timespan from 1900 to 2022. Finally, 8674 records, including articles, proceeding papers, reviews, editorial materials and letters, were obtained and a text-based file of literature information was output from the WoS database for further bibliometric analyses.

### 2.2. Data Process and Analysis

We firstly analyzed the temporal trend of records during the whole investigation period and basic information including writing language and distribution of authors, institutes, countries or journals to know the development history and contribution of researchers, institutes and countries, as well as journals that had majorly published the topic-related work.

Furthermore, the collected data were imported into CiteSpace (version 5.2.R2), a popular bibliometric software that could help us to visualize the structural and temporal patterns or development trends in a given scientific field during a given investigation period [28]. It is often used to analyze the network structures of main countries, institutions or authors and bursting terms on the basis of keywords, countries, institutions or authors. Since the number of publications was intermittent and scarce before 1980, we set up the time slicing of 1980–2022 and years per slice = 1 year for further analyses. The top 20 most cited or occurring items were selected from each slice in order to obtain the most prominent items. “Pathfinder” and “pruning the merged network” were chosen for pruning. Other parameters were set up as default.

In the analyses of network structure, the centrality of the node that reflected the importance of a node was calculated on the basis of the percentage of shortest paths in the network for a given node [28]. For co-occurrence analysis of terms, the log-likelihood ratio (LLR) was calculated to cluster the highly related terms during the period [29], with the label size representing the number of terms in each cluster. We could consider that the results of the cluster were reliable and reasonable when the silhouette index was greater than 0.5 [30]. In burst detection analysis, the length of the whole line presented the research period (1980–2022) and the red part indicated the burst period of an emerging hotspot in a specific period.

Subsequently, HistCite (Pro 2.1), a citation analysis software that could be used to analyze the history of the citation of a publication and therefore underline highly important publications in a specific period [31], was utilized to examine the prominent authors and citation relationship of the top 50 most cited records. Local citation score (LCS) and global citation score (GCS) were employed to indicate the academic impact of these publications. The former represented the citation number of a given literature by other publications in the same research field, while the latter represented the total citation number in the whole WoS database, regardless of the research fields of citations [32]. Lastly, total local citation score (TLCS) was adopted to rank the publications and authors. The current status of development and hotspots in the earthworm-related research field were determined through analyzing the citation network of highly cited papers [33].

## 3. Results and Discussion

### 3.1. Temporal Trend of Publication

The year-scale number of publications indicates the level of researchers’ interest in a given topic. As shown in this study (Figure 1), the number of publications kept increasing during the investigation period, which could be roughly divided into three stages (i.e., periods of 1900–1990, 1991–2005 and 2006–2022).

There were relatively few publications in the first period of 1900–1990. Despite that its high importance had been proposed by Darwin in 1881, earthworms did not draw much attention in the following 90 years, during which a total of 223 publications were published (on average 2.48 per year). In the earliest stage, researchers focused on the role of earthworms in shaping soil structure and fertility [34,35,36,37], and the publication record was extremely low and relatively constant. The first increase was observed in 1981, since the European Economic Community and Organization for Economic Co-operation and Development recommended earthworm as a bioindicator for the risk assessment of toxic substances in terrestrial environments in 1981 and 1984, respectively [38]. Then, researchers started to investigate the effect of chemicals on the ecotoxicology of earthworms [39,40] and were interested in their interactions with other soil organisms, such as nematodes and microorganisms [41,42].

The second stage was from 1991 to 2005, during which increasing attention was paid toward earthworm studies and the annual publication number obviously increased to range from 72 to 206. In this stage, several international symposiums with topics on earthworm ecotoxicology, soil animal science and earthworm ecology were held to encourage ecological risk assessment studies and revealed the potential of earthworms to remediate soil pollutants and improve soil quality by investigating the response of earthworms to environmental pollutants [38,43]. Related researches reported that the diversity, total number and biomass of earthworms greatly depend on the distance between the earthworms and source of heavy metals [44]. Moreover, different species of earthworms exhibited varying degrees of avoidance to heavy metal contamination due to their behavioral and ecological characteristics; for example, endogeic earthworms (*A. tuberculate*) were more sensitive to Cu and Zn contamination in the soil than epigeic earthworms (*L. rubellus*) [45]. In addition, numerous European earthworms were anthropogenically introduced to North America in this period, e.g., associated with alien plants and used as fishing bait [46,47], making the invasion of exotic earthworms to be a newly emerging research hotspot [48,49,50].

The last period ranged from 2006 to 2022 (the end point of literature searching). In this period, the number of publications increased exponentially, with more than 200 papers published every year, and numerous studies started to focus on the effects of earthworm invasion on soil structure, native soil food webs and nutrient availability, which explains the increase in the number of publications after 2006 [51,52,53]. This indicates that earthworm-related research topics constituted a bursting hotspot since then. In 2016, Earthworm Society of Britain proposed to designate 21 October of each year as the World Earthworm Day to announce the ecological value of earthworms and meanwhile commemorate Darwin as the pioneer of earthworm studies.

### 3.2. Document Types, Languages, Subject Categories and Journal Analysis

Research articles represented the majority (89.49%) of the total records in the WoS database, followed by proceeding papers (5.86%) and review papers (4.15%, Appendix A). The literatures were published in nine languages, with English being the most prevalent language, accounting for 97.71% of the records.

The publications covered 131 subject categories, with the top 10 categories being presented in Table 1. “Environmental Science” was the most popular subject category, with 37.51% of all the publications. The second was “Soil Science” (33.87%), followed by “Ecology” (17.82%). In WoS, 1013 journals were found to publish earthworm-related papers, and the top 10 journals are listed in Table 2. The most productive journal was *Soil Biology & Biochemistry*, with 612 records that accounted for 7.06% of the total. The following journals included *Applied Soil Ecology*, *Pedobiologia*, *Environmental Pollution*, *Ecotoxicology and Environmental Safety* and *Chemosphere*, which comprised 5.72%, 4.65%, 3.47%, 3.39% and 3.38% of the total publications, respectively. The top 10 journals were categoried to Environment Science, Soil Science, Ecology, Soil Zoology and Ecotoxicology, a pattern consistent with the result of subject categories.

### 3.3. Countries-, Institutions- and Authors-Based Outputs and Collaborations

#### 3.3.1. Countries-Based Outputs and Collaborations Analysis

The countries-based statistics of publication can indicate the level of research activity of countries. The top 10 countries of publication records were the United States, China, France, Germany, England, Netherlands, Spain, India, Canada and Brazil. Researchers in these countries produced 79.17% of all the records. The United States and China were major contributors that produced 15.02% and 13.67% of the total records, respectively (Table 2), indicating that they were the most active on the topic. However, the centrality indices of publications of the United States and China were not high (0.47 and 0.06, respectively; Figure 2). Because the centrality index indicates international academic status and level of influence in a given field [54], the relatively low centrality implies that a considerable percentage of publications from the two countries had not made much academic influence at the global scale. Related studies in China were relatively late to start. China and the United States had less cooperation with other countries, likely implying that the authors in the two countries preferred to cooperate with domestic researchers (Figure 2).

England had the highest level of influence in the field and highly cooperated with other countries, since both its publication record and country centrality were obviously high. England is the earliest contributor on the topic, and most of earthworm-related researches were originally conducted in England. This promoted cooperation among European countries, and many European countries such as Germany and the Netherlands also presented high influence and centrality.

#### 3.3.2. Institutions-Based Outputs and Collaborations Analysis

Analyzing institutions’ outputs can be used to assess the research capabilities and potential of an institution and identify the leading institutions on a given topic [33]. In the targeted field, the top 10 institutions constituted of 7 European and 3 Chinese institutions (Table 3). The institution with the most publications was the Chinese Academy of Sciences, which produced 317 records, and the second was the French National Institute of Agricultural Sciences (180 records), followed by the Russian Academy of Sciences, University of Chinese Academy of Sciences, Vrijie University Amsterdam, University of Gottingen, Wageningen University, China Agricultural University, University of Vigo and Aarhus University (Table 3). In spite of the high rank by the total publication record of the United States in the country-based statistics, the top 10 institutes did not cover any institute from the United States, suggesting that a large number of research institutions in the United States have been conducting earthworm-related studies.

Interestingly, the US institutes played unneglectable roles in the research topic because three United States universities with high centrality (Cornell University, University of Florida and Rice University) were ranked in the top ten list (Table 3). Therein, Cornell University had the highest centrality and made frequent cooperations with other institutions (Figure 3). Moreover, two Chinese institutions (Nankai University and Chinese Academy of Sciences) and five European institutions (the French National Institute for Agricultural Research, University of Montpellier, Center for Ecology & Hydrology, Leizig University and University of Goettingen) were listed in the top ten institutes of high centrality. Generally, European institutions had high scores in both publication record and centrality, and collaborations started earlier among these institutes than those in other countries. The observation indicates that Europe is the leading pioneer in earthworm-related research.

#### 3.3.3. Prominent Authors and Collaborations Analysis

Highly influential authors were determined by calculating TLCS, and the top 10 influential authors are shown in Table 4. Professor Lavelle P, who had 140 publication records and a TLCS score of 4817, was the most influential researcher on the topic. The second and third top authors were Scheu S and Decaens T, with TLCS scores of 3258 and 1864, respectively. Eight of the ten most influential authors came from European institutes, suggesting that European researchers had played the leading role in this field. Moreover, collaborations were frequent among researchers (Figure 4), and the highly active groups were often headed by the most prominent researchers and led wide scientific cooperation. The color of the link indicates the closeness of cooperation between authors, and we could observe that the researchers in China started the topic relatively lately and had fewer collaborations compared with European researchers.

### 3.4. Research Hotspots

The citation network in WoS was constructed based on the research objective and content of the top 50 cited papers, which were divided into four sections. The four sections represent four research hotspots, which are marked as green, yellow, red and blue, respectively (Figure 5).

The literature in the green part mainly reports the ecological effect of earthworms on soil structure, nutrient cycling, soil biota, soil ecosystem services and greenhouse gas emissions. Node 101 was the earliest highly cited paper in which Ehlers indicated the positive effect of earthworm channels on soil drainage [55]. Another paper of node 273 also emphasized the effects of earthworms on soil structure (e.g., soil aggregates, microporosity and gas permeability) [56]. Nodes 206 and 440, authored by Lavelle [42], revealed that earthworm activities significantly affected functions of the soil system and could accelerate nutrition cycling. Earthworms were introduced into agricultural systems to improve soil fertility for crop growth due to the ecological linkage between aboveground and belowground parts [57,58]. In 1995, researchers started to study the interactive effects of earthworms and soil microbes. Earthworm activities could greatly accelerate the mineralization process of soil organic matter and provide more nutrients to soil microorganisms [59]. Moreover, soil macrofauna, including earthworms, could affect nitrogen cycling processes through regulating the composition and activity of soil microbial communities [60], e.g., by supporting specific groups of the N_2_O-producing soil bacteria [61]. Then, Ingrid et al. (node 4723) made a crucial review to discuss the possibility that earthworms increased the emission of greenhouse gases [62]. The ‘Sleeping Beauty Paradox’ was proposed by Lavelle et al. to explain why earthworms changed soil properties and affected soil microbial and faunal community diversity [63,64]. These publications highlighted the importance of soil invertebrates in shaping the diversity of soil organisms. Subsequently, several crucial reviews were made to clarify the ecological effects of earthworms on soil systems from different perspectives (e.g., soil function and ecosystem services) [65,66].

The second hotspot marked in yellow (Figure 5) focuses on the effects of agricultural activities on earthworms. Conventional tillage changed the abundance and diversity of the earthworm population and resulted in significant declines in the number of earthworms [67]. Differently, permanent pasture and organic management could favor the formation of soil organic matter and earthworm population to maintain high soil fertilization and sustainable utilization of land [68]. Michel et al. suggested that earthworm activities could enhance agricultural sustainability in intensive farming systems (node 5436) [69].

Earthworm ecotoxicology occurred as the third hotspot (marked in red in Figure 5). It is well known that soil degradation and soil pollutions were further exacerbated due to mining and smelting activities of humans and the use of chemicals in agriculture, and the effects on soil biota, including earthworms, were frequently investigated [38,70]. In 1981, the earthworm was proposed to be an excellent biological indicator of soil environmental conditions. Morgan et al. first proposed that earthworms could be used to indicate soil pollutions of cadmium, copper, lead and zinc (node 208) [71] and that earthworm tissues and feces could accumulate different concentrations of metals in contaminated soil (node 1354) [72]. Spurgeon et al. further pointed out that the conventional OECD tests overestimated the impact of metals on earthworm populations (node 577) [73] because of a higher bioavailability of metals in artificial soils [74]. Subsequently, Spurgeon et al. summarized the proposals in the first three International Workshops on Earthworm Ecotoxicology and highlighted the necessity to study the ecotoxicology of earthworms (node 1961) [75]. After that, a large number of ecotoxicological studies were conducted, which stimulated the need for earthworm supply. Thus, Christopher et al. reviewed the research progress of earthworm cultivation techniques for the development of ecotoxicology and soil restoration and investigated the growth rates and reproduction of four earthworm species under laboratory conditions (node 2321) [18]. In 2009, the oxidative stress and DNA damage of earthworms were used to evaluate the effects of toxic substances on the soil environment [76]. *Eisenia fetida* was the most widely used for toxicological tests, but further studies using different species and conducted in naturally contaminated rather than artificial soils are needed before expanding the application of research results [77].

Invasion effects of alien earthworms were the fourth hotspot (marked in blue; Figure 5) because invasive earthworms were observed to dramatically affect native ecosystems in North America [48]. Invasive species, introduced into new environments, could outcompete native species for resources and habitats, thereby changing the community structure [78]. Beginning with node 658, Alban et al. reported the invasive effects of exotic earthworms on a forest soil in northern Minnesota [79]. The invasion of earthworms reduced the weight and thickness of the forest floor but increased the soil density, since more humus was produced due to the intensified mixture of organic and mineral materials by earthworm activities. Moreover, invasive earthworms could decrease the density and number of microarthropods species, alter the distribution and community composition of soil microflora and change the competition among plant species [52,80]. Earthworm invasion also significantly changed the composition of the native earthworm community [81]. Bohlen et al. proposed that the invasion of exotic earthworms played an important role in regulating soil structure and function in northern temperate forest ecosystems in the last few decades and that the existence of earthworms needed to be included in regional risk assessment [49,82]. Exotic earthworms increased the leaf C/N ratio and the effects of earthworms on soil properties depended on the total biomass and the feeding and burrowing habitats of earthworms [83]. Paul et al. proposed some endogenous and exogenous factors to determine invasive earthworm species and suggested to predict the ecological effects of invasive earthworms by combining models with molecular techniques [50]. In spite of the known effects of invasive earthworms on native ecosystems, the potential risks of earthworm invasion remain to increase with the expansion of global commerce due to a lack of legal restrictions on targeted earthworms in most parts of the world [48].

### 3.5. Term Co-Occurrence

Similar terms that are located at a research field will be clustered as a group. The structure changes in a specific domain can be revealed by conducting term co-occurrence analysis [84]. In this study, 11 clusters were established and these clusters mainly focused on four topics (Table 5).

Until the 1990s, researchers concentrated on exploring the effects of earthworms on the soil ecosystem, and this research topic covered the cluster IDs 0, 2, 4 and 8. In these clusters, researchers focused on the decomposition of leaf litter and formation of soil organic matter that were influenced by activities of earthworms. Earthworms contribute to the decomposition and transformation of leaf litter, promote soil fertility by increasing organic carbon and total nitrogen content, and enhance soil microbiota diversity [19,85]. The role of earthworms in litter decomposition is commonly long-term and lasting, but in the short term, earthworms may have no significant effect on litter decomposition [86]. Moreover, advanced techniques such as isotope labeling could be used to trace soil nutrient transfer. By a stable isotope tracer, it was found that different earthworm species had different effects in the early stage of soil organic matter mineralization, and that their burrowing activities changed the soil structure and played an important role in the stabilization of soil organic matter [56,87]. Additionally, earthworm activities were observed to impact humus profile morphology. The mineral content of the soil humus profile increased with depth due to the mixing effect of earthworms in the soil [88]. Subsequently, the interactions among earthworms and other soil animals were paid more attention and ecological roles of earthworms were further reported. For instance, researchers found that earthworm activities were able to promote plant performance, thereby indirectly improving aphid reproduction [89,90].

The second topic focused on the effects of human activities (e.g., agricultural or pedestrian activity) on earthworms, including cluster IDs 5, 6 and 7. Since intensive agricultural activities result in degradation of soil structure and increase in soil pollution, adopting new farming systems with reduced tillage and adjusted chemical input will protect the stability of the soil structure. The effects of agricultural management could be further facilitated by the presence of enchytraeids [91]. In order to protect the quality of arable land, many countries carried out policies to reduce tillage. For example, Congress of the US passed a Conservation Reserve Program to improve soil fertility, reduce soil erosion and promote the sustainable development of agriculture in 1985 [92]. After that, people paid more attention to the effects of different arable farming systems on earthworms and found that earthworm abundance and biomass were increased in the agricultural soils by reducing tillage [93,94]. The effects of reduced tillage on earthworms may be influenced by earthworm species and soil physiochemical properties. In addition, cluster ID 6 demonstrated that the effects of pedestrian activities on earthworms in forests or urban greens were concerning. Due to excessive human trampling, the soil was heavily compacted with no herb vegetation in extensive paths, resulting in decreases in both the number and diversity of earthworm populations [95,96].

Cluster IDs 1, 3 and 9 constituted the third topic, which focused on earthworm ecotoxicology. The earthworm (specifically, *Eisenia fetida*) was recommended as a biological indicator to test the effects of chemicals (such as triazole) in the soil environment by European Union, OECD and the US Environmental Protection Agency [97,98]. Moreover, earthworms could be used to remove heavy metal in the sewage sludge due to the accumulation of earthworms’ tissue and the ingestion [99].

Cluster ID 10 concentrated on the effects of exotic earthworms, which was the fourth topic. Numerous exotic earthworms invaded North America, accelerated plant litter decomposition and changed the soil organic horizon to affect native plant growth, causing forest fragmentation [100,101,102].

### 3.6. Burst Detection

The abrupt changes in entity could be revealed by conducting burst detection analyses on keywords, countries, institutions and authors of the collected literature in a certain time zone [103]. The results of burst detection analyses are shown in Table 6, Table 7, Appendix A to present the emerging topics, active countries/regions, institutions and potential authors in the targeted field in the investigated period [104].

#### 3.6.1. Emerging Hotpots

We found that the emerging research hotspots since 2018 were related with “impact (2018–2022)”, “biodiversity (2018–2019)”, “oxidative stress (2018–2022)”, “diversity (2018–2019)”, “response (2019–2022)”, “*Eisenia fetida* (2020–2022)” and “exposure (2020–2022)” (Table 6). With the growth of population and urbanization, the continuous production of waste has become a popular environmental issue. Vermicomposting, as a solution for recycling the organic component of municipal solid waste, and its impacts on ecosystems have consequently attracted more attention [105]. Earthworms could serve as indicators of soil biodiversity, quality and productivity, and therefore numerous recent studies have been conducted to elucidate the impacts of anthropogenic activities, such as the use of chemicals and changes in soil management practices, on earthworm populations [106,107,108]. It was reported that soil biodiversity was deteriorating in Europe because of the agricultural intensification and climate change [109]. The presence of earthworms changed soil biodiversity [110], and invasive earthworms were found to significantly regulate soil biodiversity [111,112] and soil microbial communities [113]. In addition, the impact of interactions between earthworms and invasive plants was concerning because of the additive effects of both [114]. Moreover, an increasing number of studies reported that the presence of earthworms could modify soil microbial community activity and composition [20], and studied the diversity of gut microbiota in earthworms [115]. The response of earthworms with exposure to chemicals or other stress conditions has attracted much attention since 2018 [116,117,118]. As known, the earthworm is an important indicator for the evaluation of ecological risk and ecotoxicology, but indications of the toxicity endpoint by observing the behavior response of earthworms are time-consuming, tedious and may be influenced by the individual variations [119]. Therefore, the response of earthworms to oxidative stress has been developed rapidly for evaluating the ecotoxicological effect of chemicals [120], in which *Eisenia fetida* has been widely chosen for the ecotoxicological assessment of chemicals due to the recommendation of international organizations [121]. In short, the ecological effects of earthworms, earthworm ecotoxicology and earthworm invasion are currently research hotspots in the topic.

#### 3.6.2. Active Countries/Regions, Institutions and Potential Authors

The top 10 active countries/regions and institutions from 1980 to 2022 were identified by the burst detection analysis (Table 7 and Appendix A). In the early stage from 1980 to 1996, the burst of country/region occurred in USA, Australia and Finland, with a burst strength of 25.76, 29.27 and 21.77, respectively (Table 7). Although earthworm-related studies started late in China, the number of publications authorized by Chinese researchers has abruptly increased between 2018 and 2022, with a strength of 123.65 (Table 7), indicating that Chinese researchers paid increasing attention to earthworm studies in the last 5 years. Moreover, the number of publications from Iran, South Africa, Austria and Japan also increased rapidly in recent years.

In the top 10 active institutions, there were 4 institutions from China, 2 from France, and 1 from each of Australia, Germany, Unites States and Netherlands (Appendix A). The bursting institutions from 1980 to 2016 were Commonwealth Scientific and Industrial Research Organization (Australia), Technical University of Darmstadt (Germany), Ohio State University (USA), Vrije University Amsterdam (Netherlands) and INRA (France). In the last five years, the burst occurred in University of Chinese Academy of Sciences (China), Chinese Academy of Sciences (China), Shandong Agricultural University (China) and Nanjing Agricultural University (China), showing that Chinese institutions played a prominent role in the period. Correspondingly, most of the bursting authors were from China in the last five years (Appendix A), and they paid much attention to revealing the impact and potential mechanisms of earthworms on the soil ecosystem.

## 4. Conclusions

In the present study, a scientometrical review was conducted by collecting and analyzing 8674 publications on the topic of earthworms in the soil ecosystem during 1900–2022. The results showed that the number of publications had been exponentially increasing since 1900 and that three stages were identified, i.e., 1900–1990, 1991–2005 and 2006–2022, with different countries leading researches on the topic in these stages. A total of 1013 journals published papers on the topic, with the most papers in *Soil Biology & Biochemistry*. The highly productive institutions were Chinese Academy of Sciences, Wageningen University, French National Institute of Agricultural Sciences, Russian Academy of Sciences and University of Chinese Academy of Sciences. At the global scale, Professor Lavelle P was the most influential author, but, in recent years, since 2018, Chinese institutions and authors had played an active role. Four main research hotspots were identified in the investigated period, i.e., effects of earthworms on the soil ecosystem, ecotoxicology of earthworms and soil remediation by earthworms, effects of alien earthworms, and effects of anthropogenic activities on earthworms. Furthermore, terms of “impact”, “biodiversity”, “oxidative stress”, “diversity”, “response”, “*Eisenia fetida*” and “exposure” had been the emerging active topics with obvious bursts since 2018, with the bursting of some terms, such as impacts, oxidative stress and exposure, lasting until the end point (2022) of searching in the present study. This observation indicates that these terms could be continuously highlighted at least in the following years. In summary, we clarified the research progress of earthworms in soil ecosystems, highly active research subjects on the topic, research hotspots in different period and emerging bursts in recent years by conducting these analyses. These results could help us to systematically understand the development progress of research on the earthworm and provide implications to allow us to know the following research hotspots at least in recent years.

## Figures and Tables

**Figure 1 biology-13-00385-f001:**
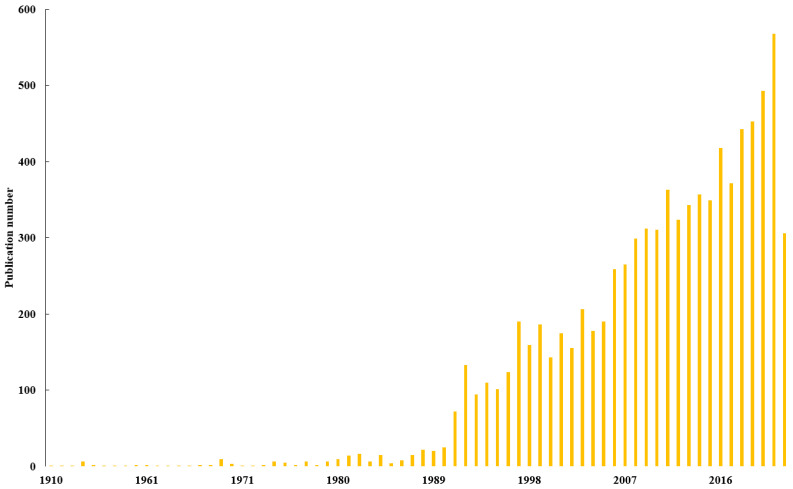
Annual publication related to earthworm research collected in WoS from 1900 to 2022.

**Figure 2 biology-13-00385-f002:**
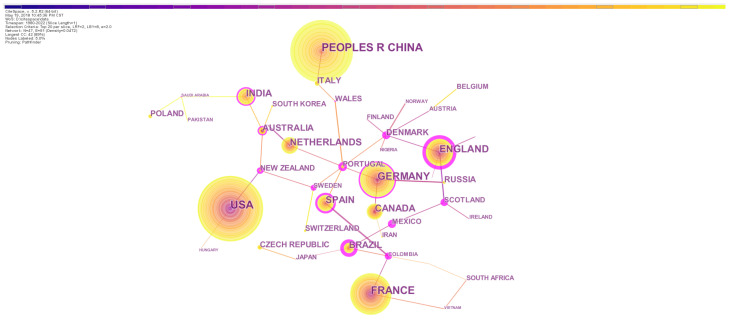
The network of country/region collaborations on earthworm-related research collected in WoS. The nodes represent countries and regions. The size is proportional to the number of countries and regions and the color of node corresponds to the year. The connections between nodes indicate cooperation between these countries/regions. Nodes with high centrality are highlighted by the purple ring.

**Figure 3 biology-13-00385-f003:**
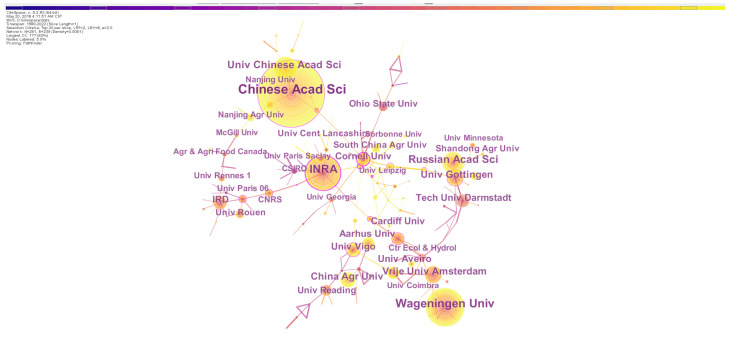
The network of institution collaborations on earthworm-related research collected in WoS. The nodes represent institutions. The size is proportional to the number of institutions and the color of node corresponds to the year. The connections between nodes indicate cooperation between these institutions. Nodes with high centrality are highlighted by the purple ring.

**Figure 4 biology-13-00385-f004:**
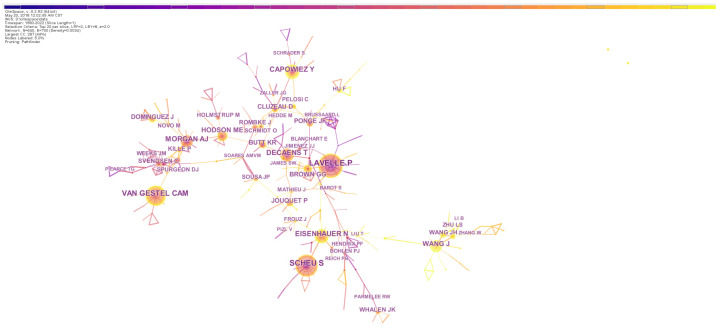
The network of author collaborations on earthworm-related research collected in WoS. The nodes represent authors. The size is proportional to the publication number of authors and the color of node corresponds to the year. The connections between nodes indicate cooperation between these authors. Nodes with high centrality are highlighted by the purple ring.

**Figure 5 biology-13-00385-f005:**
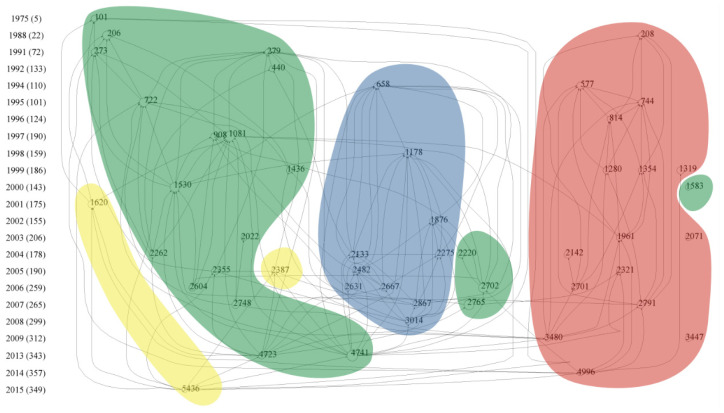
The citation network on soil earthworm-related papers collected in WoS. Each node represents a highly cited paper. If there is a link between paper A and paper B, and the arrow points to paper B, then it means that paper A cited paper B. The size of node is proportional to the LCS of paper. Note: node = 50, links = 166, LCS min = 108, max = 378 (LCS scaled).

**Table 1 biology-13-00385-t001:** Top 10 subject categories and journals on earthworm-related research collected in WoS.

Identifier	Subject Categories	Percentage/%	Journals	Percentage/%
1	Environmental Science	37.51	*Soil Biology & Biochemistry*	7.06
2	Soil Science	33.87	*Applied Soil Ecology*	5.72
3	Ecology	17.82	*Pedobiologia*	4.65
4	Toxicology	10.28	*Environmental Pollution*	3.47
5	Engineering Environmental	4.96	*Ecotoxicology and Environmental Safety*	3.39
6	Agronomy	4.53	*Chemosphere*	3.38
7	Plant Sciences	3.42	*Science of the Total Environment*	3.01
8	Agriculture Multidisciplinary	3.41	*Biology and Fertility of Soils*	2.99
9	Zoology	3.33	*European Journal of Soil Biology*	2.74
10	Biodiversity Conservation	2.88	*Environmental Toxicology and Chemistry*	2.57

**Table 2 biology-13-00385-t002:** Top 10 records and centrality based on the published papers related to earthworm research by countries/regions collected in WoS.

Identifier	Countries/Regions	Records	Countries/Regions	Centrality
1	United States	1303	Denmark	0.90
2	People’s Republic of China	1186	England	0.88
3	France	897	Brazil	0.86
4	Germany	783	Scotland	0.85
5	England	643	Colombia	0.84
6	Netherlands	473	Mexico	0.81
7	Spain	421	Sweden	0.62
8	India	419	Portugal	0.61
9	Canada	414	Spain	0.60
10	Brazil	328	New Zealand	0.53

**Table 3 biology-13-00385-t003:** Top 10 records and centrality based on the published papers related to earthworm research by institutions in WoS.

Identifier	Institutionss	Records	Institutions	Centrality
1	Chinese Acad Sci	317	Cornell Univ	0.34
2	Wageningen Univ	197	INRA	0.33
3	INRA	180	Univ Montpellier	0.26
4	Russian Acad Sci	122	Univ Florida	0.22
5	Univ Chinese Acad Sci	110	Ctr Ecol & Hydrol	0.16
6	Vrije Univ Amsterdam	90	Univ Leipzig	0.16
7	Univ Gottingen	87	Nankai Univ	0.15
8	China Agr Univ	80	Rice Univ	0.15
9	Univ Vigo	73	Chinese Acad Sci	0.14
10	Aarhus Univ	71	Univ Gottingen	0.14

**Table 4 biology-13-00385-t004:** Rank of TLCS based on the published paper records related to earthworm research by authors collected in WoS.

Identifier	Authors	Records	TLCS
1	Lavelle P	140	4817
2	Scheu S	130	3258
3	Decaens T	89	1863
4	Morgan AJ	78	1766
5	Spurgeon DJ	56	1753
6	Capowiez Y	90	1708
7	Bohlen PJ	32	1667
8	Cluzeau D	55	1577
9	Hodson ME	67	1505
10	Butt KR	66	1395

**Table 5 biology-13-00385-t005:** Top 11 clusters in terms of publications related to earthworm research fields during 1980–2022.

Identifier	Size	Silhouette	Mean (Year)	Representative Terms (LLR)
0	21	0.80	1995	^14^C-labeled beech leaf litter; sarophagous invertebrate; soil structure
1	19	0.99	2006	*Eisenia fetida*; heavy metal; sewage sludge
2	16	0.78	1996	aphid development; herbivore system; tropical sugarcane ecosystem
3	15	0.79	1999	chisel-tilled soil; earthworm community structure; triazole fungicide
4	15	0.79	1996	microbial biomass; faunal interaction; *hordelymus europaen*
5	13	0.81	1993	biogenic structure; different arable farming system; enchytraeid activity
6	12	0.88	1995	pedestrian activity; reproduction test; recycled water source
7	11	0.88	1993	microbial communities; organic arable farming; reduced tillage
8	11	0.96	1996	stabilizing principle; contrasting chemical composition; humid tropical condition
9	10	0.98	2009	soil invertebrate; tropical soil; *Eisenia andrei*
10	5	0.96	1995	exotic earthworm; laboratory condition; barley straw

**Table 6 biology-13-00385-t006:** Keywords with bursts in the publications related to earthworm research in the last five years.

Keywords	Strength	Begin	End	1980–2022
Impact	78.99	2018	2022	▂▂▂▂▂▂▂▂▂▂▂▂▂▂▂▂▂▂▂▂▂▂▂▂▂▂▂▂▂▂▂▂▂▂▂▂▂▂ ▃▃▃▃▃
Biodiversity	11.14	2018	2019	▂▂▂▂▂▂▂▂▂▂▂▂▂▂▂▂▂▂▂▂▂▂▂▂▂▂▂▂▂▂▂▂▂▂▂▂▂▂ ▃▃ ▂▂▂
Oxidative stress	59	2018	2022	▂▂▂▂▂▂▂▂▂▂▂▂▂▂▂▂▂▂▂▂▂▂▂▂▂▂▂▂▂▂▂▂▂▂▂▂▂▂ ▃▃▃▃▃
Diversity	4.06	2018	2019	▂▂▂▂▂▂▂▂▂▂▂▂▂▂▂▂▂▂▂▂▂▂▂▂▂▂▂▂▂▂▂▂▂▂▂▂▂▂ ▃▃ ▂▂▂
Response	54.31	2019	2022	▂▂▂▂▂▂▂▂▂▂▂▂▂▂▂▂▂▂▂▂▂▂▂▂▂▂▂▂▂▂▂▂▂▂▂▂▂▂▂ ▃▃▃▃
*Eisenia fetida*	23.43	2020	2022	▂▂▂▂▂▂▂▂▂▂▂▂▂▂▂▂▂▂▂▂▂▂▂▂▂▂▂▂▂▂▂▂▂▂▂▂▂▂▂▂ ▃▃▃
Exposure	53.43	2020	2022	▂▂▂▂▂▂▂▂▂▂▂▂▂▂▂▂▂▂▂▂▂▂▂▂▂▂▂▂▂▂▂▂▂▂▂▂▂▂▂▂ ▃▃▃

**Table 7 biology-13-00385-t007:** Top 10 countries/regions with bursts in terms of publication records related to earthworm research from 1980 to 2022.

Countries/Regions	Strength	Begin	End	1980–2022
People’s Republic of China	123.65	2018	2022	▂▂▂▂▂▂▂▂▂▂▂▂▂▂▂▂▂▂▂▂▂▂▂▂▂▂▂▂▂▂▂▂▂▂▂▂▂▂ ▃▃▃▃▃
Australia	29.27	1991	1999	▂▂▂▂▂▂▂▂▂▂▂ ▃▃▃▃▃▃▃▃▃ ▂▂▂▂▂▂▂▂▂▂▂▂▂▂▂▂▂▂▂▂▂▂▂
United Stated	25.76	1980	1996	▃▃▃▃▃▃▃▃▃▃▃▃▃▃▃▃▃ ▂▂▂▂▂▂▂▂▂▂▂▂▂▂▂▂▂▂▂▂▂▂▂▂▂▂
Scotland	24.55	1996	2009	▂▂▂▂▂▂▂▂▂▂▂▂▂▂▂▂ ▃▃▃▃▃▃▃▃▃▃▃▃▃▃ ▂▂▂▂▂▂▂▂▂▂▂▂▂
Finland	21.77	1990	2005	▂▂▂▂▂▂▂▂▂▂ ▃▃▃▃▃▃▃▃▃▃▃▃▃▃▃▃ ▂▂▂▂▂▂▂▂▂▂▂▂▂▂▂▂▂
Iran	14.51	2020	2022	▂▂▂▂▂▂▂▂▂▂▂▂▂▂▂▂▂▂▂▂▂▂▂▂▂▂▂▂▂▂▂▂▂▂▂▂▂▂▂▂ ▃▃▃
South Africa	13.09	2007	2012	▂▂▂▂▂▂▂▂▂▂▂▂▂▂▂▂▂▂▂▂▂▂▂▂▂▂▂ ▃▃▃▃▃▃ ▂▂▂▂▂▂▂▂▂▂
Austria	12.24	2017	2019	▂▂▂▂▂▂▂▂▂▂▂▂▂▂▂▂▂▂▂▂▂▂▂▂▂▂▂▂▂▂▂▂▂▂▂▂▂ ▃▃▃ ▂▂▂
England	12.07	1998	2005	▂▂▂▂▂▂▂▂▂▂▂▂▂▂▂▂▂▂ ▃▃▃▃▃▃▃▃ ▂▂▂▂▂▂▂▂▂▂▂▂▂▂▂▂▂
Japan	11.64	2012	2015	▂▂▂▂▂▂▂▂▂▂▂▂▂▂▂▂▂▂▂▂▂▂▂▂▂▂▂▂▂▂▂▂ ▃▃▃▃ ▂▂▂▂▂▂▂

## Data Availability

The data that support the finding of this study are available from the corresponding author upon reasonable request.

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
