# Peer review of "A Bibliometric Analysis on Research Progress of Earthworms in Soil Ecosystems"

_biology, 2024, doi:10.3390/biology13060385_

Round 1

Reviewer 1 Report (New Reviewer)

Comments and Suggestions for Authors

Comments for Author,

Thank you for your submission. I have read your paper carefully, unfortunately, I can not suggest for publicationbecause these below shortness

1- There are a lot of problem the design of bibliometric analyses paper.

2- Lack of enough novel content and connection between subtitle.

3- There is any real discussion part in the paper with subtitles.

4- The methods are unclear. Please add properly methods to improve this part.

Best Regards

Author Response

Review1: Thank you for your submission. I have read your paper carefully, unfortunately, I can not suggest for publication because these below shortness.

1- There are a lot of problem the design of bibliometric analyses paper.

Response: Thank you for your time to evaluate our work. In this manuscript, we aimed to clarify the research progress of the topic of earthworm in soil ecosystem by using the bibliometric method. Therefore, we searched literature in the database of Web of Science by topic of ‘soil’ plus ‘earthworm’, which could provide the most publications as possible on the topic. The literature data were further analyzed in CiteSpace (for network relationship of research agents [countries, institutes or researchers], burst points, and co-occurrence analysis of terms at different stages) and in HistCite to identify the prominent authors and citation network of top cited papers. This literature collection strategy is suitable on basis of our expectation and the methods and software used here are correct in a bibliometric paper.

2- Lack of enough novel content and connection between subtitle.

Response: We agree with you that a bibliometric analysis is not highly novel as experimental studies, because it provide information of research progress of a given topic to readers. It is more likely the summary of details in a research topic, such as distribution, relationships and cooperation of authors, institute or countries. Although it is not novel enough, this type of study could tell us the development stages and important information such as prominent authors and leading institutes, as well as cooperation and citation network. The information are important to researchers, especially those newcomers such as undergraduates and freshman of postgraduate. In a bibliometric study, a subsection generally reports an aspect of results of the bibliometric analysis and therefore shows weak connectivity, which is a bit different from experimental studies.

3- There is any real discussion part in the paper with subtitles.

Response: We agree with you that the manuscript is weak on discussion. As suggested, we strengthened discussion on the reasons of these observations, such as different development stages of research, cooccurrence of terms and burst detection analysis, in this revision (lines 173-179, 185-187, 370-376, 381-382, 388-404,427-436, 438-440, 444-449).

4- The methods are unclear. Please add properly methods to improve this part.

Response: As suggested, we added more details in the methodology to clarify how we conducted this study and which methods were used to obtain the results (line 110-118 and line 126-139).

Reviewer 2 Report (Previous Reviewer 1)

Comments and Suggestions for Authors

I carefully read the feedback from the other two reviewers and improved the manuscript according to their comments. The manuscript has become even better. As I checked the revised manuscript very carefully and read it slowly, I found some minor typos, which I list below.

Line 100 - extra dot

Lines 127-129 “while the latter represented the total citation number in the whole WoS database, regardless that the research fields of citations was the same ot  not” – OR NOT

Line 143 [33,34][35-37]? May be so: [33,34, 35-37]

Line 143 languages,subject categories - missing gap

Line 245 of the top 50 cited papers , - extra gap

Line 334 IDs 0, 2, 4 and 8.In these  - missing gap

Line 337 differentearthworm  - missing gap

Line 386 earthworms could modify the modification - Modification is probably an unnecessary word

Author Response

Review2: I carefully read the feedback from the other two reviewers and improved the manuscript according to their comments. The manuscript has become even better.

Response: Thank you to review our manuscript and make the positive comment on our effort.

As I checked the revised manuscript very carefully and read it slowly, I found some minor typos, which I list below.

Line 100 - extra dot

Lines 127-129 “while the latter represented the total citation number in the whole WoS database, regardless that the research fields of citations was the same ot not” – OR NOT

Line 143 [33,34][35-37]? May be so: [33,34, 35-37]

Line 143 languages, subject categories - missing gap

Line 245 of the top 50 cited papers , - extra gap

Line 334 IDs 0, 2, 4 and 8.In these - missing gap

Line 337 differentearthworm - missing gap

Line 386 earthworms could modify the modification - Modification is probably an unnecessary word

Response: Sorry for the typos and format errors. Following your suggestion, we fixed all these problems (lines 158-159, 193, 277 and 367) and carefully proofread the manuscript (lines 143 and 441) in this revision.

Round 2

Reviewer 1 Report (New Reviewer)

Comments and Suggestions for Authors

Author made all deserved revision, now the paper could be accepted for the publication.

Best Regards

This manuscript is a resubmission of an earlier submission. The following is a list of the peer review reports and author responses from that submission.

Round 1

Reviewer 1 Report

Comments and Suggestions for Authors

This literature review analyses scientific studies on earthworms published between 1900 and 2022. According to the authors, interest in this topic has grown particularly since 2006. The review is written on Web of Science (WoS) Core Collection databases on 18 October 2022 (8674 articles, proceeding papers, reviews, editorial materials). The authors identified three periods in terms of the intensity of publications on the topic: 1900-1990, 1991-2005, 2006-2022. Analyses of publications on the topic in various journals, institutions, institutions and authors outputs and collaborations have been carried out. The hottest topics discussed are highlighted. Soil Biology & Biochemistry is the most popular journal for publishing relative results. Four institutions from China, two from France and one each from Australia, Germany, the United States and the Netherlands were among the top 10 most active institutions. Chinese institutions and authors have been most active in this topic between 2018 and 2022. The review is short and concise. The review can be useful for further study of the scientific topic of soil earthworms, selecting the most relevant direction of research.

Reviewer 2 Report

Comments and Suggestions for Authors

I was not convinced that this study will help scientists to better identify the hotspots and trends in the future. Where do the authors see continued work and why?

The title includes "current" and "future" perspectives. However, it really only analyzes the "past."

More detail should be given about how network statistics are calculated (TLCS, GCS, Centrality) so that readers can better know how to interpret the results.

Many of the tables and figures might be eliminated. Much of what is in them is redundant with the existing text.

Currently, the "Conclusions" section is more-or-less just a summary (redundant with the abstract). I was looking for a true, synthetic conclusion based on this analysis of the literature. In what direction is earthworm research moving and why?

Comments on the Quality of English Language

The English needs much work. 

Citation style is highly non-conventional (e.g., use of first names, reference to "Author et al." without a year following, etc.).

Reviewer 3 Report

Comments and Suggestions for Authors

The manuscript is well-written and easy to read and follow. 

However, there are some points that I would stress more:

- improve the aims; the last paragraph of the introduction is a summary of what was done and which answers were answered, however it is not clear why this work was important and which will be the contribution to the field (besides being a review of the papers regarding earthworms).

- at page 9, line 176, the authors introduced the concept of invasion, exotic earthworms, invasive earthworms, etc and their effects on the community without explaining what an invasive species is or which is the difference with an endemic species, etc. The term invasive was then substituted to alien earthworms along the text, but I would actually stick to invasive species. Not all the non-indigenous species (NIS) become invasive ones, and I would pay more attention on how you use the term invasive or exotic. I am not even sure that exotic species is a scientific term usually in most of the recent literature. Please add what missing and adjust accordingly.

- line 72-73, citations in the text are incomplete, please add the year.

- line 106, be precise when writing about "several questions"

- line 114, merge the second sentence with the previous one

- line 324, substitute "they" with the authors
